# Effects of various living-low and training-high modes with distinct training prescriptions on sea-level performance: A network meta-analysis

**Xinmiao Feng, Yonghui Chen, Teishuai Yan, Hongyuan Lu, Chuangang Wang, Linin Zhao** *

Sports Coaching College, Beijing Sport University, Haidian, Beijing, China

* fengxinmiao666@163.com

**Data Availability Statement:** All relevant data are within the manuscript and its Supporting Information files.

## Abstract

This study aimed to separately compare and rank the effect of various living-low and training-high (LLTH) modes on aerobic and anaerobic performances in athletes, focusing on training intensity, modality, and volume, through network meta-analysis. We systematically searched PubMed, Web of Science, Embase, EBSCO, and Cochrane from their inception date to June 30, 2023. Based on the hypoxic training modality and the intensity and duration of work intervals, LLTH was divided into intermittent hypoxic exposure, continuous hypoxic training, repeated sprint training in hypoxia (RSH; work interval: 5–10 s and rest interval: approximately 30 s), interval sprint training in hypoxia (ISH; work interval: 15–30 s), short-duration high-intensity interval training (s-IHT; short work interval: 1–2 min), long-duration high-intensity interval training (l-IHT; long work interval: > 5 min), and continuous and interval training under hypoxia. A meta-analysis was conducted to determine the standardized mean differences (SMDs) among the effects of various hypoxic interventions on aerobic and anaerobic performances. From 2,072 originally identified titles, 56 studies were included in the analysis. The pooled data from 53 studies showed that only l-IHT (SMDs: 0.78 [95% credible interval; CrI, 0.52–1.05]) and RSH (SMDs: 0.30 [95% CrI, 0.10–0.50]) compared with normoxic training effectively improved athletes' aerobic performance. Furthermore, the pooled data from 29 studies revealed that active intermittent hypoxic training compared with normoxic training can effectively improve anaerobic performance, with SMDs ranging from 0.97 (95% CrI, 0.12–1.81) for l-IHT to 0.32 (95% CrI, 0.05–0.59) for RSH. When adopting a program for LLTH, sufficient duration and work intensity intervals are key to achieving optimal improvements in athletes' overall performance, regardless of the potential improvement in aerobic or anaerobic performance. Nevertheless, it is essential to acknowledge that this study incorporated merely one study on the improvement of anaerobic performance by l-IHT, undermining the credibility of the results. Accordingly, more related studies are needed in the future to provide evidence-based support. It seems difficult to achieve beneficial adaptive changes in performance with intermittent passive hypoxic exposure and continuous low-intensity hypoxic training.

**Funding:** The author(s) received no specific funding for this work.

**Competing interests:** The authors have declared that no competing interests exist.

## Introduction

The hypoxic training techniques of "Live High-Train High" (LHTH) and "Live High-Train Low" (LHTL), which necessitate athletes to spend daily prolonged durations in high altitudes, have been used for over half of a century [1, 2]. This camp-style hypoxic technique is often employed by individual and team-sport athletes during the pre-season training phase to gain a competitive edge. However, research has suggested that such long-term hypoxic exposure training can potentially cause a range of detrimental effects in athletes (such as muscle cell deterioration and immunological disruptions etc.) [3–8]. In recent years, with the evolution of hypoxia equipment and to mitigate the drawbacks associated with chronic hypoxia [9], most studies have focused on LLTH modes that expose athletes to only discrete hypoxia during specific training or rest periods [10–12]. Recent reviews have introduced updated panorama for hypoxia training [13, 14] and divided LLTH into two major categories (active and passive) based on the parameters of the training regimen. The passive hypoxic paradigm is intermittent hypoxic exposure (IHE), where athletes are exposed to short (<3 hours) yet intense passive hypoxia during rest periods [15]. The active hypoxic paradigm encompasses continuous hypoxia training (CHT), which is a moderate-intensity training that lasts for approximately 30 minutes [16], and intermittent (moderate-high intensity) hypoxia training. According to the training duration and ratio of work to rest, intermittent training can be divided into repeated sprint training in hypoxia (RSH; work interval: 5–10 s and rest interval: approximately 30 s) [17], interval sprint training in hypoxia (ISH; work interval: 15–30 s) [18], high-intensity interval training (IHT; short duration: 1–2 min [19] and long duration: > 5 min [20]). In addition, some studies have combined continuous and intermittent hypoxia training aimed at optimizing aerobic and anaerobic capabilities concurrently [21–23].

Current literature has suggested that LLTH variants cannot trigger adequate hypoxic stimuli to induce related hematological alterations [24–26]. The associated primary mechanism is to induce muscular and peripheral adaptations [15, 27] by adding extra hypoxic load during training. However, the ongoing discourse about the effectiveness of various LLTH variants remains unresolved [28–31]. Critics argue that restriction of oxygen availability could substantially decrease absolute training intensity [31, 32]. This could potentially induce degenerative changes, subsequently limiting the enhancement of sea-level performance. The first meta-analysis conducted by Bonetti and Hopkins (2009) on hypoxic training suggested that compared with normoxic training, LLTH does not significantly enhance athletes' performance. However, some perspectives have highlighted that previous studies did not consider the potential impact of variable training prescriptions [10, 14]. Recent meta-analyses have also suggested that high-intensity interval training [33] or repeated sprint training [34] in a hypoxic condition can effectively improve performance or aerobic capacity in athletes. Training at different duration and intensities has been shown to place diverse physiological demands on the body, leading to potential differences in subsequent adaptive changes, irrespective of whether the environment is normoxic or hypoxic [28, 35]. Hence, discerning the disparities in performance outcomes across diverse training modalities of LLTH is crucial for its practical application.

Compared with the usual meta-analysis used in previous studies, network meta-analysis (NMA) can generate a clear hierarchy between various interventions by synthesizing the results of direct and indirect comparisons to derive more comprehensive and definitive comparison results [36, 37]. Therefore, LLTH variants were meticulously divided based on their various training parameters. Subsequently, we conducted an NMA to comprehensively compare and rank the performance (i.e., aerobic and anaerobic) enhancement effects of various hypoxic training modes in athletes.

## Methods

This meta-analysis was conducted in accordance with the Preferred Reporting Items for Systematic Reviews and Meta-Analyses NMA (S1 Checklist and S1 File) [38]. Additionally, the study protocol has been registered with PROSPERO under registration number CRD42023421683.

### Data source and search

The following English electronic databases were searched systematically: PubMed, Web of Science, Embase, EBSCO, and Cochrane from their inception date to June 30, 2023. The following search syntax was utilized: ("altitude" OR "hypoxia") AND ("high-intensity interval training" OR "repeated sprint training" OR "interval sprint training" OR "continuous training" OR "intermittent exposure" OR "performance") (S1 Table). we manually searched all review articles related to altitude or hypoxic training and traced additional possible studies by reviewing their reference lists.

### Study selection

The standards were based on the PICOS approach (participants, interventions, comparators, outcomes, and study design). The participants were athletes who had received training but had not been exposed to hypoxia in the last 6 months. There were no criteria regarding the sport type and training level of the participants (youth teams, university teams, national teams, professional club teams, etc.). We categorized LLTH into seven modes based on the training type, intensity, and volume (duration of work interval) (for detailed classification and definition, see Fig 1 and Table 1 in S2 File) (Table 1). All the training plans for the control group are implemented in a normoxic environment. In the face-to-face studies included in the selected systematic reviews, the comparator should be any of the seven hypoxic modes. At least one test result of interest should have been obtained in the study. The results were mainly divided into

**Table 1. Definition of LLTH modes.**

| Hypoxic training mode | Definition |
|---|---|
| Repeated sprint training in hypoxia (RSH) | the repetition of several short "all-out" exercise bouts (≤15 s) in hypoxia interspersed with incomplete recoveries (30 s, exercise-to-rest ratio <1:4) |
| Interval sprint training in hypoxia (ISH) | Several "all-out" sprints bouts (usually 30 s) in hypoxia interspersed with recoveries (2–5 min) |
| Short-duration high-intensity Interval training (s-IHT) | Several short-term high-intensity exercise bouts (1–3 min) in hypoxia interspersed with recoveries (2-5min) |
| Long-duration high-intensity interval training (l-IHT) | Several long-term high-intensity exercise bouts (>3 min) in hypoxia interspersed with recoveries (2-5min) |
| Continuous hypoxic training (CHT) | Moderate-high intensity continuous training (30-60min) in hypoxia |
| Continuous and Interval training under Hypoxia (C+I) | 1) One session consisting of continuous and interval training 2) Interval and continuous training sessions were conducted separately during a week |
| Intermittent Hypoxic exposure (IHE) | intermittent exposure to a severe hypoxia during rest 1) Alternatively receiving normoxia and hypoxia exposure 2) Persistently receiving hypoxia exposure |

two categories according to the metabolic characteristics in the test: aerobic and anaerobic performances. The reference indicators for aerobic and anaerobic performances include some test results that have been proven to be highly correlated. We have chosen the Incremental Treadmill Test (Time to exhaustion (min) etc.), 3 min All-Out Test (average power output (W) etc.), YYIR test level I and II (Ddistance Covered (m)), Run test (distance ≥ 1000 m; duration (seconds or minutes)), Cycle test (durations ≥ 10 min; duration (seconds or minutes)), Swiming test (distance ≥ 400 m; duration (seconds or minutes)) and constant-load test (Time to exhaustion (seconds or minutes) etc.) as the reference indicatorfor aerobic performance, and the Wingate Test (average power output (W) etc.), RAS test (peak power output (W)), Maximal anaerobic test (duration (seconds or minutes)), Supramaximal time to fatigue test (duration (seconds or minutes)), Run test (distance ≤ 800 m, or durations ≤ 2 min; duration (seconds or minutes)) and Swiming test (≤ 200 m; duration (seconds or minutes)) as the standard for anaerobic performance, for details on the selection of reference indicators, S4 File. In designing this study, we mainly considered randomized controlled studies (including the front and back parts of crossover randomized controlled studies) and conducted a meta-network analysis of studies on various types of hypoxic and normoxic training. We excluded studies on a single exercise in an acute hypoxia condition. In addition, studies that included special training interventions (i.e., cold, hot, or humid environments, blood flow restriction intervention, etc.) or special nutritional supplements (i.e., nitrate, caffeine, etc.) were excluded.

## Data extraction and quality evaluation

All relevant articles retrieved from the aforementioned electronic databases were stored in EndNote X9 reference manager, and three investigators reviewed and selected the retrieved articles based on the aforementioned reference criteria. Subsequently, relevant data was extracted from the qualified articles. Information extracted included publication information (author and year), research participants (sample size, gender ratio, participants' sports, and training level), experimental design (type of experiment), intervention measures (hypoxia mode and hypoxia dosage [km/h]; details of the hypoxic dose model are provided in S3 File), duration and frequency of training (training plan and supplementary training), and test results (the outcome measures selected per study are shown in the Table 2). The hypoxic dosage between different hypoxic types was coordinated using the "kilometer hours" model [39]. The dosage model was defined as km·h = (m/1000)× h ("m" represents the altitude of the exposure environment; "h" represents the total exposure duration).

The methodological quality of the Included articles was evaluated by two independent reviewers using the Physiotherapy Evidence Database (PEDro) scale [40]. The PEDro scale includes 11 items made up of three items from the Jadad scale and nine items from the Delphi list. The PEDro scale score for randomized controlled trials ranges from 0 (low quality) to 10 (high quality), and a score of ≥6 represents high-quality research. The first item on the PEDro scale (eligibility criteria specified) is used to establish external validity; thus, the score is not included in the total score. Any disagreement during the above process was resolved by a review group within the team (HY, YH, TS and XM) through consensus and arbitration.

## Statistical analysis

We used the R software (version 3.6.3) package netmeta (www.rproject.org) to perform the NMA, combining direct and indirect comparisons in a frequency model [41, 42]. The standardized mean difference (SMD) and 95% credible interval (CI) was used as effect size indicators. We used the random effects NMA model to collate the size of the study effect. The circle size in the network evidence graph represents the sample size, and the lines between the circles

**Table 2. Characteristics of included studies.**

| study | design | participants sample size | participants sports | hypoxic protocol | intervention length and frequency (weeks × sessions per week; total days/total sessions) | training protocol (sets numbers of works interval number×duration of intensity separated by rest interval duration one session remained duration (of intensity) | height (m) | testing mode aerobic | testing mode index | testing mode anaerobic | testing mode index | Score |
|---|---|---|---|---|---|---|---|---|---|---|---|---|
| S.R. Goods et al. 2015 | RCT-S | N:6m H:5m | football | RSH | 5 × 3 | 3 sets of 7 × 5-s sprints separated by 15-35s | 3000 | | | run RSA test | PPO(W) | 7 |
| Giovanna et al. 2022 | RCT-N | N:9m H:10m | endurance | RSH | 2 × 3 | 4 sets of 5 × 10-s cycling sprints separated by 20s | 4000 | incremental treadmill test 250 kJ time trial | PPO(W) Duration (s) | maximal anaerobic cycling test | Duration (s) | 7 |
| Kasai et al. 2015 | RCT-S | N:16f H:16f | lacrosse | RSH | 2 × 4 | 2 sets of 10 × 7-s cycling sprints separated by 30s | 3000 | incremental treadmill test | Time to exhaustion (s) | cycling RSA test | PPO(W) | 7 |
| Wadee et al. 2022 | RCT-S | N:7m H:7m | Rugby | RSH | 6 × 3 | 3 sets of 10 × 6-s sprints separated by 30s | 3000 | incremental treadmill test | Time to exhaustion (s) | run RSA test | PPO(W) | 8 |
| Montero et al. 2016 | C | N:8m H:7m | endurance | RSH | 4 × 3 | 3 sets of 5 × 10-s cycling sprints separated by 20s | 3000 | incremental treadmill test | Time to exhaustion (s) | run RSA test | PPO(W) | 8 |
| Faiss et al. 2013 | RCT-S | N:20m H:20m | cycling | RSH | 4 × 2 | 3 sets of 5 × 10-s cycling sprints separated by 20s | 3000 | 3 min All-Out Test | APO(W) | cycling RSA test Wingate Test | PPO(W) APO(W) | 5 |
| M Galvinet al. 2013 | RCT-S | N:15m H:15m | rugby | RSH | 4 × 3 | 10 × 6-s cycling sprints separated by 30s | 3500 | YYIR test level 1 | distance covered (m) | | | 5 |
| Gatterer et al. 2014 | RCT-S | N:5m H:5m | football | RSH | 5 × 1.6 | 3 sets of 5 × 10-s cycling sprints separated by 20s | 3300 | YYIR test level 2 | distance covered (m) | run RSA test | PPO(W) | 8 |
| Faiss et al. 2015 | RCT-D | N:6m 3f H:5m 3f | ski | RSH | 2 × 3 | 4 sets of 5 × 10-s cycling sprints separated by 20s | 3000 | | | run RSA test Single 10-s sprint | PPO(W) APO(W) | 8 |
| Brocherie et al. 2015 | RCT-D | N:12m H:12m | field hockey | RSH | 2 × 3 | 4 sets of 5 × 5-s cycling sprints separated by 25s | 3000 | YYIR test level 2 | distance covered (m) | | | 7 |
| Brocherie et al. 2015 | RCT-D | N:8m H:8m | football | RSH | 5 × 2 | 2–3 sets of 5–6 × 15-s running sprints separated by 30s | 2900 | incremental Field test | maximal aerobic speed (km·h-1) | 40m time trial run RSA test | sprinting times(s) PPO(W) | 7 |
| Brechbuhl et al. 2020 | RCT-D | N:11m H:11m | tennis | RSH | 12 days/5 sessions | 4 sets of 4 × 8-s running sprints separated by 30s | 3000 | incremental treadmill test | Time to exhaustion (s) | run RSA test | best sprinting times(s) | 8 |

*(Continued)*

**Table 2.** (Continued)

| study | design | participants sample size | participants sports | hypoxic protocol | intervention (length and frequency; weeks × sessions per week; total days/total sessions) | training protocol (sets numbers of works interval number×duration of intensity separated by rest interval duration one session remained duration (of intensity)) | height (m) | testing mode aerobic | testing mode index | testing mode anaerobic | testing mode index | Score |
|---|---|---|---|---|---|---|---|---|---|---|---|---|
| Kasaiet al. 2017 | RCT-S | N:9m H:10m | track and field | RSH | 5 days/10 sessions | two sessions per day (morning, afternoon) morning session: ①3 sets of 5 × 6-s running sprints separated by 24s ②4 × 20-s running sprint separated by 5-15s afternoon session: ①3 sets of 5 × 6-s running sprints separated by 24s ②4 × 20-s running sprint separated by 5-15s | 3000 | incremental treadmill test | Time to exhaustion (s) | Wingate Test | PPO(W); APO(W) | 7 |
| Wang et al. 2018 | RCT-D | N:9m H:8m | endurance | RSH | 4 × 2 | 3 sets of 5 × 10-s running sprints separated by 20s | 2900 | incremental treadmill test | PPO(W) | run RSA test | PPO(W) | 7 |
| Shi et al. 2023 | RCT-S | N:12m Ha:10m Hb:10 | team-sprot | RSH | Ha: 2 × 3 Ha: 5 × 3 | 3 sets of 5 × 5-s running sprints separated by 25s | 3500 | YYIR test level 1 | distance covered (m) | run RSA test | PPO(W) | 4 |
| Gatterer et al. 2018 | RCT-D | RSH:6m IHT:5m | team-sprot | RSH ISH | 3 × 3 | RSH: 3 sets of 5 × 10-s running sprints separated by 20s ISH: 4 × 30-s separated by 5min | 2200 | YYIR test level 2 | distance covered (m) | cycling/run RSA test Wingate Test | best sprinting times(s) APO(W) PPO(W) APO(W) | 5 |
| Warnier et al. 2020 | RCT-S | N:8m; Ha:8m Hb:7m Hc:7m | endurance | ISH | 6 × 2 | 4–9 repetitions of 30s running sprints separated by 4.5min | Ha: 2000 Hb: 3000 Hc: 4000 | incremental treadmill test Time trial (600 kJ) | PPO(W) Time trial (s) | Wingate Test | PPO(W) APO(W) | 7 |
| Karabiyik et al. 2021 | RCT-S | Na:8m; Nb:8m Ha:8m; Hb:8m | team-sprot | ISH | short-ISH: 4 × 3 long-ISH: 4 × 3 | Ha: 4–6 repetitions of 15s running sprints separated by 4min Hb: 4–6 repetitions of 30s running sprint separated by 4min | 3536 | incremental treadmill test | Time to exhaustion (s) | Wingate Test | PPO(W) APO(W) | 6 |

(Continued)

**Table 2.** (Continued)

| study | design | participants sample size | participants sports | hypoxic protocol | intervention (length and frequency (weeks × sessions per week; total days/total sessions)) | training protocol (sets numbers of works interval number×duration of intensity separated by rest interval duration one session remained duration (of intensity)) | height (m) | testing mode aerobic | testing mode index | testing mode anaerobic | testing mode index | Score |
|---|---|---|---|---|---|---|---|---|---|---|---|---|
| Truijens et al. 2002 | RCT-D | N:3m 5f H:3m 5f | swimming | s-IHT | 5 × 3 | one session included 3 sets ①10 bouts × 30-s separated by 15s ②5 bouts × 60-s separated by 30s ③5 bouts × 30-s separated by 15s (each sets separated by 5min) | 2500 | 400m swim | Time trial (s) | 100m swim | Time trial (s) | 7 |
| Arezzolo et al. 2020 | RCT-S | N:9m H:9m | bicycling | s-IHT | 8 × 2 | 8 × 1min of 120% VO2peak separated by 5min | 3000 | incremental treadmill test | PPO(W) | Supramaximal time to fatigue test | Time to exhaustion (s) | 6 |
| Ambrozy et al. 2020 | RCT-N | N:15m H:15m | boxing | s-IHT | 6 × 2 | 1st–4th week: 8 × 10s of 100% HRmax separated by 50s 5th–6th week: 3 sets of 5 × 20s of 100% HRmax separated by 3min | 4000 | incremental treadmill test | Maximal speed (km/h) | Wingate test | PPO(W) APO(W) | 5 |
| Morton et al. 2005 | RCT-N | N:8m H:8m | team-sport | s-IHT | 4 × 3 | 10 × 1min of 80–90% Peak Power Output separated by 2min | 2750 | incremental treadmill test | PPO(W) | Wingate test | PPO(W) APO(W) | 5 |
| Roels et al. 2005 | RCT-N | N:8m H:10m | bicycling and triathlon | CHT s-IHT | 7 × 2 | CHT: one session remained 60min of 50% VO2max IHIT: ①2 sets of 3–4 × 2min separated by 2min (6 sessions) ②1–2 sets of 3–4 × 3min of 100% PPO separated by 3min (4 sessions) ③4 × 5-8min of 90% PPO separated by 3-4min (3 sessions) | 3000 | incremental treadmill test 10min cycle time trial | PPO(W) APO(W) | | | 5 |
| Holliss et al. 2014 | RCT-S | N:7m H:5m | distance running | CHT | 8 × 2 | one session remained 30min of speed to lactate turnpoint | 2150 | incremental treadmill test | Time to exhaustion (s) | | | 4 |
| Hendriksen et al. 2003 | RCT-N | N:5m 6f H:6m 6f | triathlon | CHT | 10 days/10 sessions | one session of 60–70% heart rate reserve remained 75-105min | 2500 | incremental treadmill test | PPO(W) | Wingate Test | PPO(W) APO(W) | 7 |

(*Continued*)

Table 2. (Continued)

| study | design | sample size | sports | hypoxic protocol | intervention length and frequency (weeks × sessions per week; total days/total sessions) | training protocol (sets numbers of works interval number×duration of intensity separated by rest interval duration one session remained duration (of intensity)) | height (m) | aerobic | index | anaerobic | index | Score |
|---|---|---|---|---|---|---|---|---|---|---|---|---|
| Ponsot et al. 2005 | RCT-N | N:7m H:8m | distance running | 1-IHT | 6 × 2 | week 1–2: 2 × 12min of vVT2 separated by 5min week 3–4: 2 × 16min of vVT2 separated by 5min week 5–6: 2 × 20min of vVT2 separated by 5min | 3000 | constant-load test | Time to exhaustion (s) | | | 6 |
| Zoll et al. 2005 | RCT-D | N:6m H:9m | distance running | 1-IHT | 6 × 2 | week 1–2: 2 × 12min of vVT2 separated by 5min week 3–4: 2 × 16min of vVT2 separated by 5min week 5–6: 2 × 20min of vVT2 separated by 5min | 3000 | incremental treadmill test | Time to exhaustion (s) | | | 5 |
| Czuba et al. 2017 | RCT-N | N:7m H:8m | swimming | 1-IHT | 4 × 2 | a circuit of 90% VO2peak remained 45-55min | 2500 | incremental treadmill test | PPO(W) | Wingate Test | PPO(W) APO(W) | 6 |
| Park et al. 2022 | RCT-N | N:10f H:10f | distance running | 1-IHT | 6 × 3 | 10 × 5min of 90–95% HRmax separated by 1min | 3000 | 3000m running | Time trial (s) | | | 5 |
| Morris et al. 2020 | RCT-S | N:10m H:10m | cycling | 1-IHT | 3 × 3 | 4 × 4–5 min of 105–115% of maximal steady state heart rate separated by 10min | 1850 | incremental treadmill test | maximal steady state PPO(W) | | | 5 |
| Jung et al. 2020 | UCT | N:10m H:10m | middle- and long-distance running | 1-IHT | 6 × 3 | 10 × 5min of 90–95% HRmax separated by 1min | 3000 | 3000m running | Time trial (s) | | | 6 |
| Millet et al. 2013 | RCT-N | N:6m H:6m | basketball | 1-IHT | 3 × 3 | 4–5 × 4min 90% of vVO2max separated by 4min | 2500 | incremental treadmill test | Total distance (m) PPO(W) | | | 5 |
| Czuba et al. 2011 | RCT-N | N:10m H:10m | cyclists | 1-IHT | 3 × 3 | one microcycle remained 30-40min of 95% lactate threshold workload | 2250 | incremental treadmill test 30km cycling | PPO(W) Time trial (s) | | | 6 |
| Czuba et al. 2018 | RCT-N | N:10m H:10m | cyclists | 1-IHT | 4 × 3 | one session remained 30-40min of 100% lactate threshold workload | 2100 | incremental treadmill test 30km cycling | PPO(W) Time trial (s) | | | 6 |

(Continued)

Table 2. (Continued)

| study | design | participants sample size | participants sports | hypoxic protocol | intervention length and frequency (weeks × sessions per week; total days/total sessions) | training protocol (sets numbers of works interval number×duration of intensity separated by rest interval duration one session remained duration (of intensity) | height (m) | testing mode aerobic | testing mode index | testing mode anaerobic | testing mode index | Score |
|---|---|---|---|---|---|---|---|---|---|---|---|---|
| | | | | | | | | | | | | 7 |
| Czuba et al. 2019 | RCT-S | N:7m H:7m | biathletes | 1-IHT | 3 × 3 | week 1–2: 2 × 12 min of vVT2 separated by 2 min week 3–4: 2 × 16 min of vVT2 separated by 2 min week 5–6: 2 × 20min of vVT2 separated by 2min | 2000 | incremental treadmill test | PPO(W) | | | 6 |
| Dufour et al. 2005 | RCT-N | N:9m H:9m | distance running | 1-IHT | 6 × 2 | week 1–2: 2 × 12min of 90% HRmax separated by 2min week 3–4: 2 × 16min of 90% HRmax separated by 2min week 5–6: 2 × 20min of 90% HRmaxT2 separated by 2min | 3000 | incremental treadmill test | Time to exhaustion (s) | | | 6 |
| Sanchez et al. 2018 | RCT-D | N:6m H:9m | endurance | 1-IHT | 6 × 3 | 2 × 5 min of 80% vVO2max separated by 5 min | 5000 | incremental treadmill test | Time to exhaustion (s) Maximal aerobic speed (km·h-1) | | | 5 |
| Ramos-Campo et al. 2015 | RCT-U | N:9m H:9m | triathlon | C+I | 7 × 2 | continuous:1 session remained 60min of 60–70% PPO (11 sessions) interval:1 session remained 60 min of intensity over the anaerobic threshold (3 sessions) | 2750 | incremental treadmill test | Time to exhaustion (s) | | | 5 |
| Roels et al. 2007 | RCT-N | N:9m H:10m | cycling and triathlon | C+I | 3 × 5 (2 interval and 3 continuous training sessions per week) | CHT: one session remained 60min of 60% VO2max IHT:2 sets of 3 × 2min of 100% VO2max separated by 2 min | 3000 | incremental treadmill test 10min cycle time trial | PPO(W) APO(W) | | | 6 |

(Continued)

Table 2. (Continued)

| study | design | participants sample size | participants sports | hypoxic protocol | intervention (weeks × sessions per week; total days/total sessions) | training protocol (sets numbers of works interval number×duration of intensity separated by rest interval duration one session remained duration (of intensity)) | height (m) | testing mode aerobic | testing mode index | testing mode anaerobic | testing mode index | Score |
|---|---|---|---|---|---|---|---|---|---|---|---|---|
| | | | | | | | | | | | | 7 |
| Millet et al. 2014 | RCT-N | N:8m H:10m | cycling | C+I | 3 × 2 | CHT: one session remained 60 min of 60% VO2max IHT: 2 sets of 3 × 2 min 100% PPO separated by 6 min | 3000 | incremental treadmill test | PPO(W) | 2min all-out exercise test | APO(W) | 6 |
| Kim et al. 2021 | UCT | N:10m H:10m | swimming | C+I | 6 × 3 | CHT: one session of 75% HRmax remained 30min IHT: 10 × 2min of 90% HRmax separated by 1min | 3000 | 400m swiming | Time trial (s) | | | 7 |
| Robach et al. 2014 | RCT-D | N:8m H:9m | endurance | C+I | 6 × 3 | CHT: one session of 60% PPO remianed 60min IHT: one session of 65–130% PPO remianed 60min | 2500 | incremental treadmill test constant-load test | PPO(W) Total time (s) APO(W) | | | 7 |
| Hamlin et al. 2010 | RCT-S | N:6m 1f H:8m 1f | cycling | C+I | 5 days/5 sessions | CHT: one session of 60–70% of HR reserve remained 90 min IHT: 2 × 5 min separated by 5 min recovery | 4100 | 30km cycling | APO(W) | Wingate test | PPO(W) APO(W) | 4 |
| Park et al. 2018 | CT | N:5m 5f H:5m 5f | swimming | C+I | 6 × 3 | CHT: one session of 80% HRmax remained 30 min IHT: 10 × 2 min of 90% HRmax separated by 1 min recovery | 3000 | 400m swiming | Time trial (s) | | | 8 |
| Julian et al. 2003 | RCT-D | N:7m 1f H:7m 1f | distance running | IHE | 4 × 5 | one session remained 70 min (5:5-min normobaric and hypoxia exposure at rest) | 3000 | 3000m running | Time trial (s) | | | 5 |
| Hinckson et al. 2006 | RCT-D | N:1m 4f H:2m 5f | rowing | IHE | 3 × 7 | one session remained 90 min (alternating 6 min hypoxic and 4 min normobaric) | 3000 | 5000m running | Time trial (s) | | | 8 |
| Katayama et al. 2004 | RCT-N | N:7m H:8m | endurance | IHE | 2 × 7 | one session remained 3 hour | 4000 | 3000m running | Time trial (s) | | | 7 |

(Continued)

**Table 2.** (Continued)

| study | design | sample size | sports | hypoxic protocol | intervention length and frequency (weeks × sessions per week; total days/total sessions) | training protocol (sets numbers of works) interval number×duration of intensity separated by rest interval duration one session remained duration (of intensity) | height (m) | aerobic | index | anaerobic | index | Score |
|---|---|---|---|---|---|---|---|---|---|---|---|---|
| Rodríguez et al. 2014 | RCT-D | N:12m H:11m | swimming | IHE | 4 × 5 | one session remained 3 hour | 4750 | 400m swimming 3000m running | Time trial (s) Time trial (s) | 100 swimming | Time trial (s) | 7 |
| Tadibi et al. 2007 | RCT-D | N:10m H:10m | endurance | IHE | 15 days/15 sessions | one session remained 1 hour (hypoxia to 6 min and normoxia to 4 min) | 5550 | incremental treadmill test | PPO(W) | Wingate Test | PPO(W) | 8 |
| Miller et al. 2014 | RCT-D | N:4m H:4m | swimming | IHE | 3 × 3 | one session remained 1.5 hour | 2300 | | | 100 swimming | Time trial (s) | 8 |
| Gough et al. 2019 | RCT-D | N:3m 6f H:2m 5f | triathlon | IHE | 17 days/17 sessions | one session remained 1 hour (hypoxia to 6min and normoxia to 4 min) | 3000 | incremental treadmill test | Time to exhaustion (s) | | | 8 |
| Burtsche et al. 2010 | RCT-D | N:3m 2f H:5m 1f | middle-distance running | IHE | 5 × 3 | one session remained 2 hour | 4000 | incremental treadmill test | Time to exhaustion (s) | | | 6 |
| Katayama et al. 2003 | RCT-D | N:3m 5f H:3m 5f | swimming | IHE | 3 × 3 | one session remained 1.5 hour | 4500 | 3000m running | Time trial (s) | | | 7 |
| Bonetti et al. 2006 | C | N:5m H:5m | kayak paddlers | IHE | 3 × 5 | one session remained 1hour | 6875 | incremental treadmill test | PPO(W) | simulated 500-m race on the kayak ergometer; Single 10-s sprint | PPO(W) APO(W) | 6 |
| Lázaro et al. 2002 | RCT-S | N:12m H:12m | middle- and long-distance running | IHE | 8 × 7 | one session remained 1.5hour | 4750 | 1000m running | Time trial (s) | 60m running 400m running | Time trial (s) | 7 |
| Hamlin et al. 2002 | RCT-S | N:8m 2f H:5m 7f | endurance | IHE | 3 × 5 | one session remained 45min | 4700 | 3000m running | Time trial (s) | | | 6 |

RCT/DB "randomized double blind controlled experiment"; RCT/SB "randomized single blind control experimen"; RCT/UB "randomized unblind controlled experiment"; C "Crossover experiment"; UCT "unrandomized control experiment"; f "female"; m "male"; IHE "intermittent hypoxic exposure"; CHT "continuous hypoxic training"; RSH "Repeated sprint training in hypoxia"; ISH "Interval sprint training in hypoxia"; s-IHT "Short-trem high-intensity Interval training"; l-IHT "Long-term high-intensity interval training"; C+I "Continuous and Interval training under Hypoxia"; CON "control group (normoxic training)"; PPO "peak power output"; APO "average power output"; RSA "repeated sprint ability"; YYIR test "The Yo-Yo Intermittent Recovery Test"; N "normoxic group", H "hypoxic group".

represent direct comparisons between two types of physical activities or interventions. The width of the connecting line reflects the number of studies that directly compared the two interventions. If no connecting line is established between the two types of physical activities, we performed indirect comparisons using NMA. In addition, the SMD and 95% CI of all paired comparisons are reported, and the effect of each physical activity intervention on maximal oxygen uptake compared with that observed in the control group is reported in a league table using a forest plot. We ranked the types of physical activities using P scores based on the degree of improvement in maximal oxygen uptake among athletes. The P score ranges from 0 to 1, with a higher score indicating a greater improvement in aerobic capacity [43]. We used the tau squared ($\tau^2$) test and p-value to qualitatively analyze heterogeneity between the studies [44, 45]. The larger the $\tau^2$ and the smaller the p-value, the bigger the heterogeneity. Conversely, the smaller the $\tau^2$ and the bigger the p-value, the smaller the heterogeneity. In addition, we used $I^2$, which is distributed between 0% and 100%, to quantitatively analyze the heterogeneity between the study results. An $I^2$ less than 25% indicated low heterogeneity, ranging from 25% to 50% indicated medium heterogeneity; and >75% indicated high heterogeneity. Therefore, when $I^2$ was >50%, the heterogeneity was significant. We used global and local methods to test for inconsistency in the study results. For global inconsistency, we used the design-by-treatment test to evaluate inconsistency [46]. In addition, we used the node-splitting method in the R netmeta package for the local inconsistency test [47]. We conducted network meta-regression analysis using the R gemtc package to analyze potential sources of heterogeneity (publication year, sample size, mean age, percent male, exercise duration, exercise frequency, and total time per session). We compared adjusted funnel plots to evaluate the risk of publication bias under specific circumstances. Additionally, we analyzed the funnel plots using the Egger method. A p<0.05 indicated publication bias. We evaluated the sensitivity of our study by repeating each NMA after excluding studies with high risk of bias.

## Results

### Study characteristics and quality assessment

The search process of the systematic reviews is shown in Fig 1. After excluding 2,292 reports based on the title and abstract, 444 full-text articles were retrieved. During the examination of the full texts, we selected and included 56 studies with interesting results (the citations included studies are provided in S3 Table). A total of 1,040 participants, most of whom were male (n = 964/92.69%), were included in the 56 studies. Additionally, most participants were endurance athletes (n = 770/74.04%). All the participants in the included studies, except two who were boxing and tennis players, were team-sports athletes (n = 270/25.96%). The sample size for the studies ranged from 4 to 25. The training period ranged from 5 to 56 days (average, 28.9 days, standard deviation [SD] = 13.214), and the weekly exercise training frequency ranged from 2 to 7 (average frequency, 3.44, SD = 1.67). The specific parameters of the training programs in each study are shown in Table 2 (the scoring details per study are provided in S2 Table). The PEDro scale was used to determine the quality of the included study, with results showing an average score of 6.327±1.203 and indicating a generally high methodological quality. Only three studies had scores below the predetermined threshold of 5 points.

### Network meta-analysis

**Aerobic performance.** Fig 2 displays a network diagram of the qualified aerobic performance comparisons; all hypoxic training methods were compared with normoxic training at least once. A total of 53 studies reported changes in their primary outcome, aerobic performance among 1,021 participants (98.17%). The forest plot (Fig 3) shows that only l-IHT and

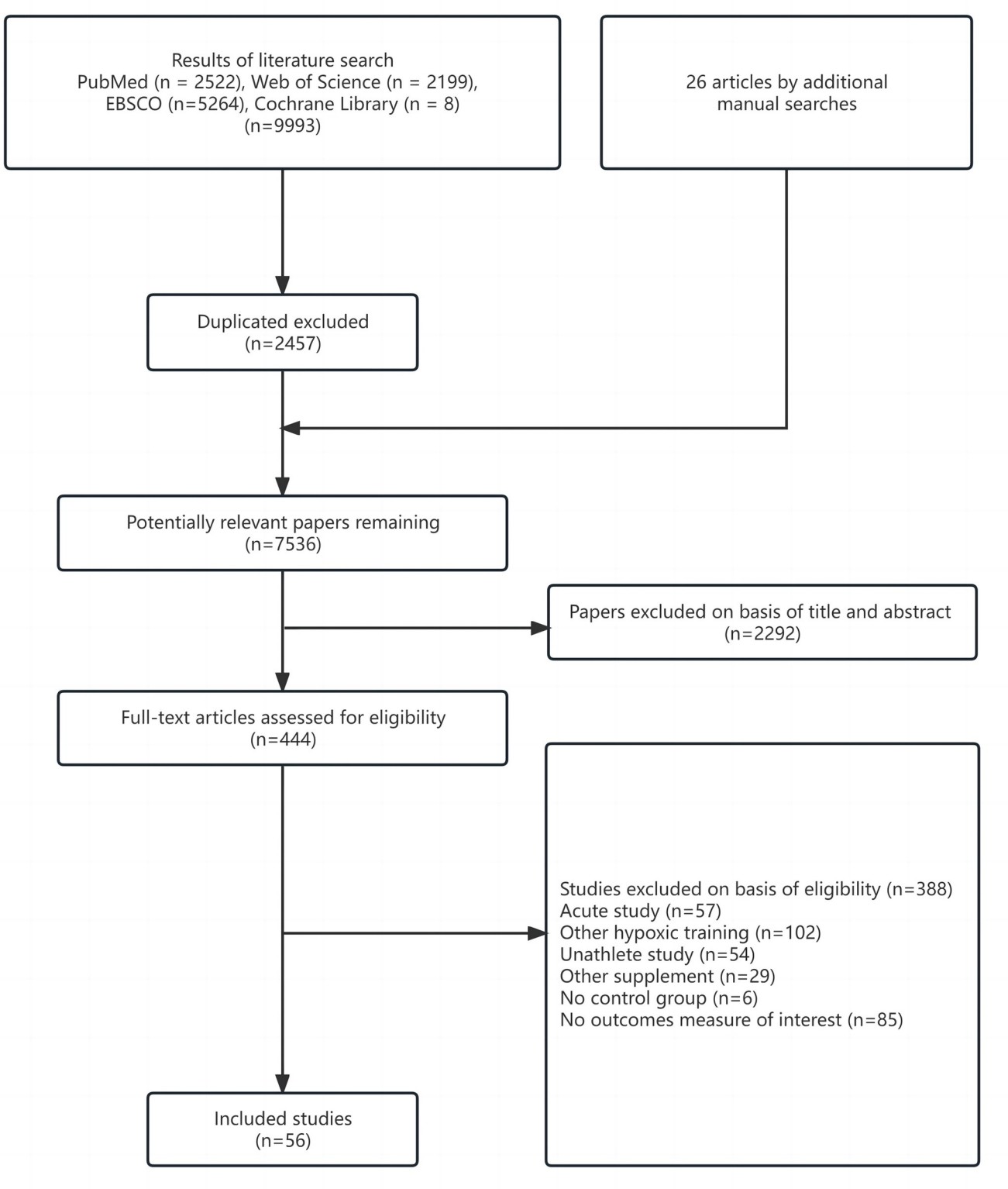

**Fig 1. Search terms and outcomes.**

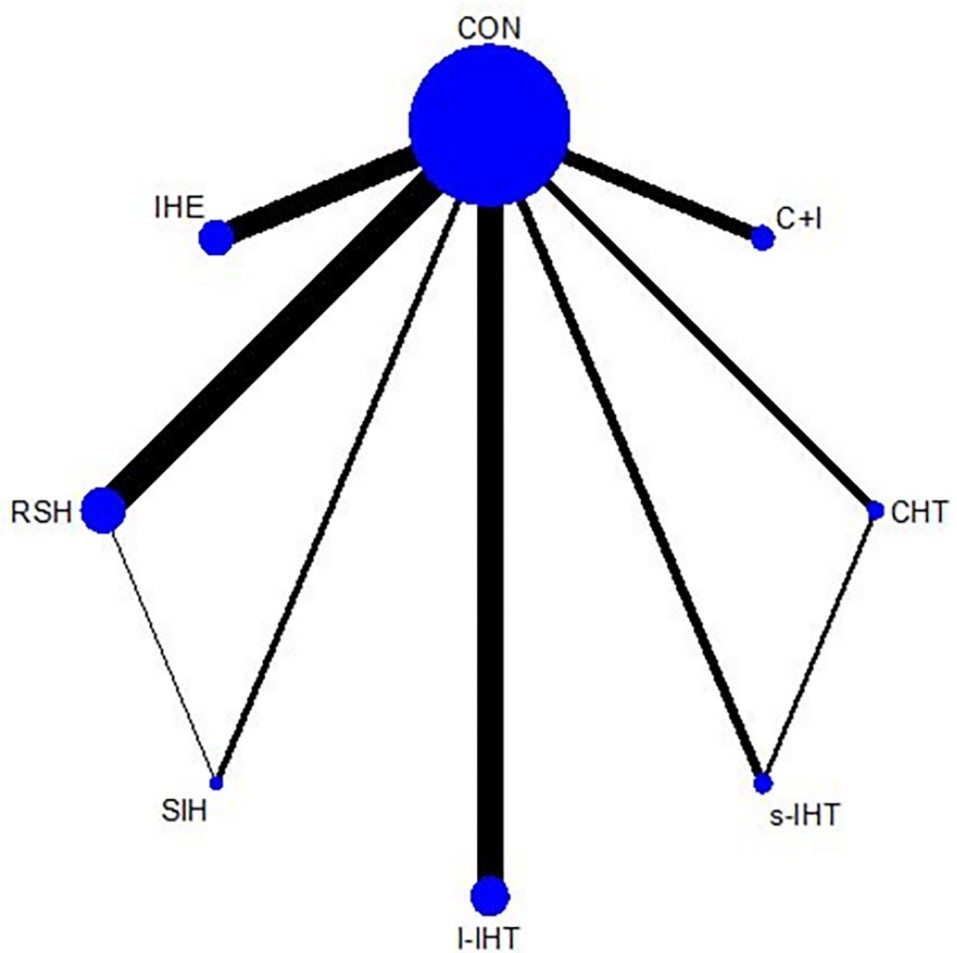

**Fig 2. Network plot of aerobic performance.** The size of the nodes corresponds to the number of participants randomized to each hypoxic training. Exercise type with direct comparisons are linked with a line; its thickness corresponds to the number of trials evaluating the comparison. IHE "intermittent hypoxic exposure"; CHT "continuous hypoxic training"; RSH "Repeated sprint training in hypoxia"; ISH "Interval sprint training in hypoxia"; s-IHT "Short-trem high-intensity Interval training"; l-IHT "Long-term high-intensity interval training"; C+I "Continuous and Interval training under Hypoxia"; CON "control group (normoxic training)".

RSH were significantly more effective than normoxic training in improving athletes' aerobic performance, with SMDs of 0.78 (95% credible interval [CrI], 0.52–1.05) for l-IHT and 0.30 (95% CrI, 0.10–0.50) for RSH. Based on the P scores, l-IHT had the best effect (P score: 1.00). The league plot (Table 3) results showed that l-IHT can improve an athlete's aerobic performance better than all the other hypoxic modes, with an SMD ranging from 0.49 and 0.78. The $I^2$ value for aerobic performance was 8% (low heterogeneity). The global Q score for inconsistency was 0.55 with a p-value of 0.6774 (Statistical methods in details, evaluation of heterogeneity and inconsistency in S5 and S6 Files).

## Anaerobic performance

Fig 4 shows the network graph of anaerobic performance comparisons, where only RSH and ISH were directly compared. 29 studies reported changes in anaerobic performance in the main results of 568 participants (54.62%). Compared with conventional oxygen training, all intermittent hypoxia training methods (l-IHT, ISH, s-IHT, and RSH) showed significant

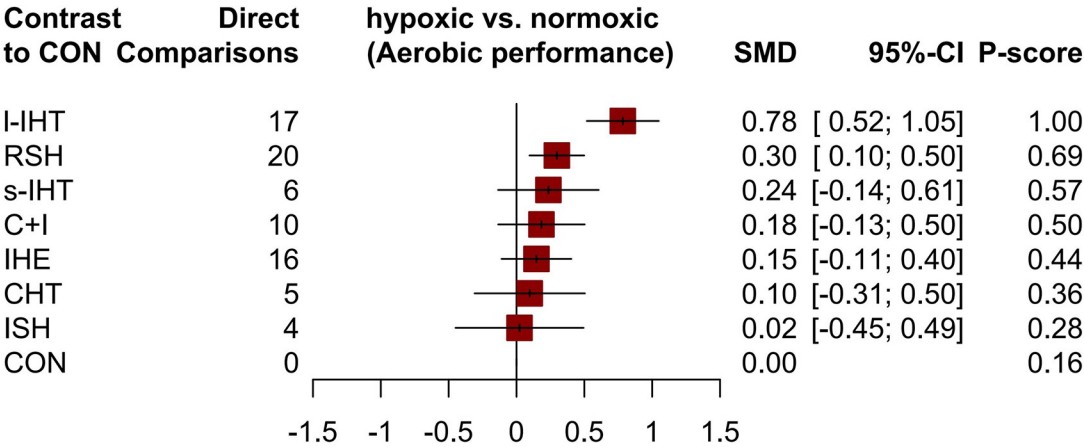

**Fig 3. Forest plot change in effect of aerobic performance.** Various LLTH modes are ranked according to the surface under the curved cumulative ranking probabilities. Treatments crossing the y-axis are not signifcantly diferent from CON. The n value represents the number of studies that were directly compared to the control group. SMD "standardized Mean Diference"; CrI "Credible Interval"; IHE "intermittent hypoxic exposure"; CHT "continuous hypoxic training"; RSH "Repeated sprint training in hypoxia"; ISH "Interval sprint training in hypoxia"; s-IHT "Short-trem high-intensity Interval training"; l-IHT "Long-term high-intensity interval training"; C+I "Continuous and Interval training under Hypoxia"; CON "control group".

improvements in anaerobic performance, with SMDs (95% CrI) ranging from 0.97 (0.12–1.81) for l-IHT to 0.32 (0.05–0.59) for RSH, and l-IHT ranks first with a P-score of 0.95 (Fig 5). In addition, the NMA results (Table 4) showed that all intermittent hypoxic training could improve anaerobic performance better than continuous and interval training under hypoxia and normoxic training. However, it should be noted that this study only included one study

**Table 3. League table for changes in aerobic performance associated with various LLTH modes.**

| l-IHT | . | . | . | . | . | . | **0.78 (0.52;1.05)** |
|---|---|---|---|---|---|---|---|
| 0.49 (0.15;0.82) | RSH | . | . | . | . | -0.03 (-1.25;1.19) | **0.31 (0.10;0.51)** |
| 0.55 (0.09;1.00) | 0.06 (-0.36;0.48) | s-IHT | . | . | 0.18 (-0.49;0.85) | . | 0.20 (-0.18;0.58) |
| 0.60 (0.19;1.01) | 0.11 (-0.26;0.49) | 0.05 (-0.44;0.54) | C+I | . | . | . | 0.18 (-0.13;0.50) |
| 0.64 (0.27;1.01) | 0.15 (-0.18;0.48) | 0.09 (-0.36;0.54) | 0.04 (-0.37;0.4) | IHE | . | . | 0.15 (-0.11;0.40) |
| 0.69 (0.20;1.17) | 0.20 (-0.25;0.65) | 0.14 (-0.35;0.62) | 0.09 (-0.43;0.6) | 0.05 (-0.43;0.53) | CHT | . | 0.07 (-0.36;0.49) |
| 0.76 (0.22;1.30) | 0.28 (-0.23;0.78) | 0.21 (-0.39;0.81) | 0.16 (-0.41;0.73) | 0.12 (-0.41;0.66) | 0.08 (-0.55;0.7) | ISH | -0.03 (-0.54;0.48) |
| 0.78 (0.52;1.05) | 0.30 (0.10;0.50) | 0.24 (-0.14;0.61) | 0.18 (-0.13;0.5) | 0.15 (-0.11;0.40) | 0.10 (-0.31;0.5) | 0.02 (-0.45;0.49) | CON |

All results are presented in the form of SMD (95% CrI). various LLTH modes are ranked according to the surface under the curve cumulative for overall effect starting with the best from left to right. The results of the network meta-analysis are showed in the lower left part, and results from pairwise comparisons in the upper right half (if available). Cells shown in bold indicate signifcant results. IHE "intermittent hypoxic exposure"; CHT "continuous hypoxic training"; RSH "Repeated sprint training in hypoxia"; ISH "Interval sprint training in hypoxia"; s-IHT "Short-trem high-intensity Interval training"; l-IHT "Long-term high-intensity interval training"; C+I "Continuous and Interval training under Hypoxia"; CON "control group (normoxic training)".

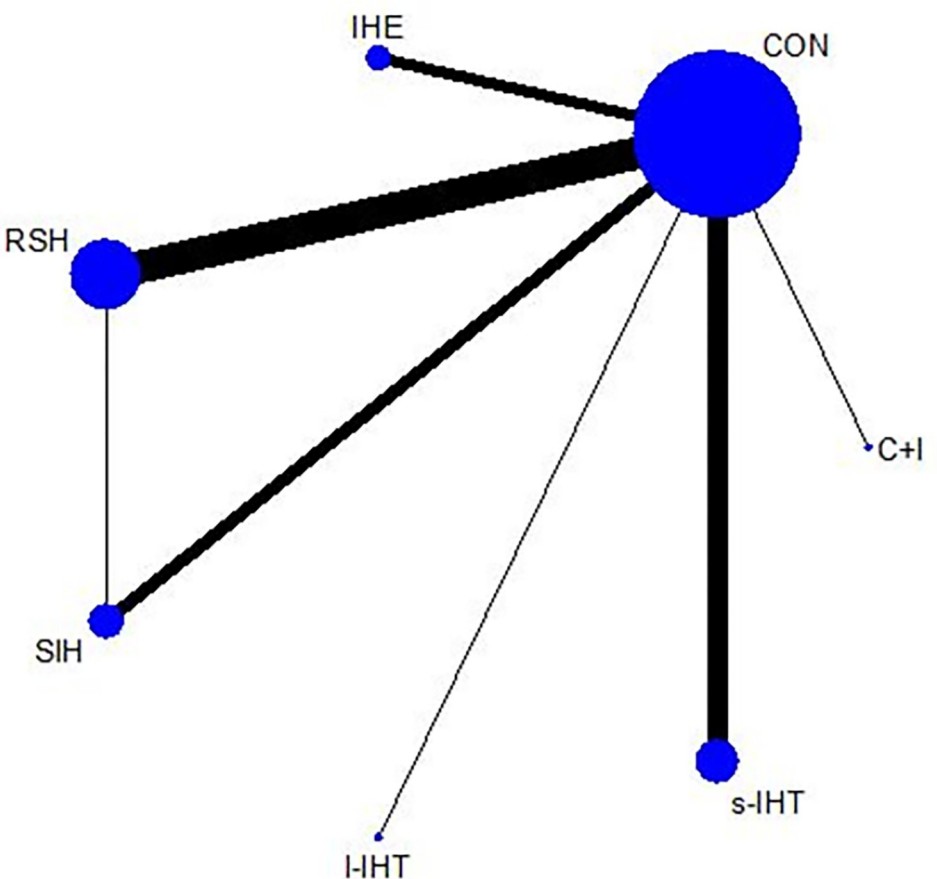

**Fig 4. Network plot of anaerobic performance.** The size of the nodes corresponds to the number of participants randomized to each hypoxic training. Exercise type with direct comparisons are linked with a line; its thickness corresponds to the number of trials evaluating the comparison. IHE "intermittent hypoxic exposure"; CHT "continuous hypoxic training"; RSH "Repeated sprint training in hypoxia"; ISH "Interval sprint training in hypoxia"; s-IHT "Short-trem high-intensity Interval training"; l-IHT "Long-term high-intensity interval training"; C+I "Continuous and Interval training under Hypoxia"; CON "control group (normoxic training)".

on the improvement of anaerobic performance by l-IHT. The $I^2$ value for anaerobic performance was 20.5% (low heterogeneity). The global Q score for inconsistency was 0.14 with a p-value of 0.7037 (Statistical methods in details, evaluation of heterogeneity and inconsistency in S5 and S6 Files).

## Discussion

This study classified LLTH into several specialized hypoxic modes according to the type, intensity, and volume of training prescription and subsequently used NMA to comprehensively compare and rank the effects of the various hypoxic modes on athletes' aerobic and anaerobic performances. The results showed that only active intermittent hypoxic modes (l-IHT, s-IHT, RSH, and ISH) compared with normoxic training were effective in improving athletes' performance. Of these, for both aerobic and anaerobic performances, l-IHT with a high volume (longer duration of training interval) and intensity showed the best results. It seems difficult to achieve beneficial adaptive changes in performance with intermittent passive hypoxic exposure and continuous low-intensity hypoxic training.

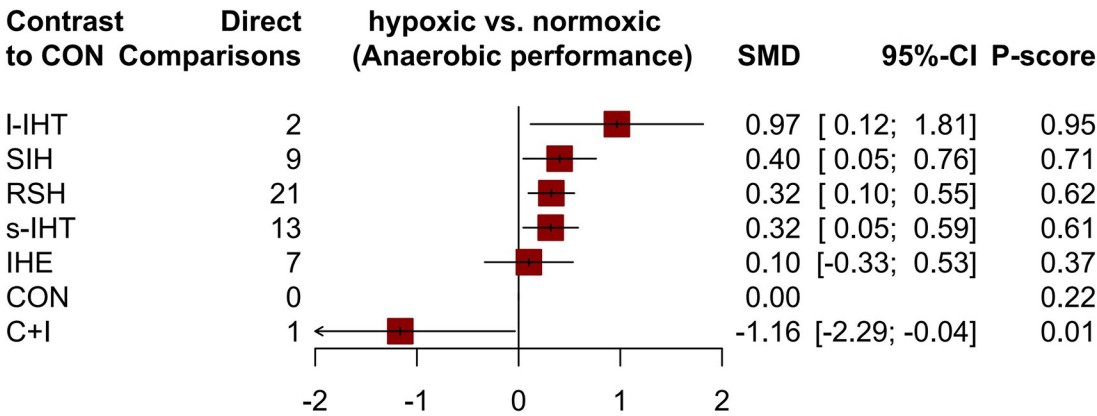

**Fig 5. Forest plot change in effect of anaerobic performance.** Various LLTH modes are ranked according to the surface under the curved cumulative ranking probabilities. Treatments crossing the y-axis are not signifcantly diferent from CON. The n value represents the number of studies that were directly compared to the control group. SMD "standardized Mean Diference"; CrI "Credible Interval"; IHE "intermittent hypoxic exposure"; CHT "continuous hypoxic training"; RSH "Repeated sprint training in hypoxia"; ISH "Interval sprint training in hypoxia"; s-IHT "Short-trem high-intensity Interval training"; l-IHT "Long-term high-intensity interval training"; C+I "Continuous and Interval training under Hypoxia"; CON "control group (normoxic training)".

The pooled data indicates that among various LLTH modes, only intermittent modes with high-intensity and large-volume training had significant large effects on enhancing aerobic performance. The findings substantiate the perspective that high intensity is integral to enhancing an athlete's endurance during hypoxic training [10, 15, 35]. In a hypoxic environment, the diminished oxygen content inevitably affects arterial oxygen pressure ($SaO_2$), reducing oxygen availability to muscle and brain tissues and constraining aerobic metabolic function [48]. Such physiological responses could notably impair performances during training with high aerobic components [49–51]. Additionally, it has been confirmed that high-level

**Table 4. League table for changes in anaerobic performance associated with various LLTH modes.**

| l-IHT | . | . | . | . | . | **0.97 (0.12;1.81)** |
|---|---|---|---|---|---|---|
| 0.56 (-0.36;1.48) | ISH | . | 0.32 (-0.97;1.61) | . | . | 0.38 (0.01;0.76) |
| 0.65 (-0.24;1.54) | 0.09 (-0.36;0.54) | s-IHT | . | . | . | 0.32 (0.05;0.59) |
| 0.64 (-0.23;1.52) | 0.08 (-0.33;0.50) | -0.01 (-0.36;0.35) | RSH | . | . | 0.32 (0.10;0.56) |
| 0.87 (-0.09;1.82) | 0.30 (-0.26;0.87) | 0.21 (-0.30;0.73) | 0.22 (-0.27;0.71) | IHE | . | 0.10 (-0.33;0.53) |
| 2.13 (0.72;3.54) | 1.57 (0.38;2.75) | 1.48 (0.32;2.64) | 1.49 (0.37;2.63) | 1.26 (0.06;2.47) | C+I | -1.16 (-2.29;-0.04) |
| 0.97 (0.12;1.81) | 0.40 (0.04;0.76) | 0.32 (0.05;0.59) | 0.32 (0.10;0.55) | 0.10 (-0.33;0.53) | -1.16 (-2.29;-0.04) | CON |

All results are presented in the form of SMD (95% CrI). various LLTH modes are ranked according to the surface under the curve cumulative for overall effect starting with the best from left to right. The results of the network meta-analysis are showed in the lower left part, and results from pairwise comparisons in the upper right half (if available). Cells shown in bold indicate signifcant results. IHE "intermittent hypoxic exposure"; CHT "continuous hypoxic training"; RSH "Repeated sprint training in hypoxia"; ISH "Interval sprint training in hypoxia"; s-IHT "Short-trem high-intensity Interval training"; l-IHT "Long-term high-intensity interval training"; C+I "Continuous and Interval training under Hypoxia"; CON "control group (normoxic training)".

athletes are likely to encounter greater impairments [52, 53]. Absolute exercise intensity in the hypoxic group that is not only markedly lower than the normoxic group but also falls short of the threshold that could invoke beneficial adaptations is observed when CHT, requiring a relatively lesser intensity, is implemented [10, 54]. Conversely, matching a higher training intensity (VT2/anaerobic threshold/80% VO$_2$max) during intermittent hypoxia training can mobilize a larger proportion of anaerobic metabolism to participate in energy supply [55], limiting the decrease in absolute exercise intensity [56]. Presently, it is understood that CHT exhibits a strong correlation with the amplification of aerobic performance and metabolic adaptations in active muscles. Hypoxic training with adequate intensity may enhance muscle oxidative capacity by optimizing substrate selection and augmenting mitochondrial function [57–59]. As high-intensity training continues, a large amount of lactic acid gradually accumulates in the tissues, which strongly stimulates the buffering capacity of the muscles [27]. Previous reports have indicated that after high-intensity hypoxic training, compared with normoxic training, the monocarboxylate transporter (MCT-1/4) related to H+ management shows significant adaptive changes [23, 60, 61], and the density of the capillaries in the working muscles (the ratio of muscle fibers to capillaries) also increases significantly [54, 62].

In addition to high intensity, longer work-interval duration (higher volume of training and hypoxic dose) is closely related to the improvement of endurance and aerobic capability [63–66]. Our results support this hypothesis, as both ISH and s-IHT did not show superiority over normoxic training. Adequate training volume cannot only pose greater challenge to aerobic metabolism and pH regulation but can also trigger special adaptations with more hypoxic dose application. A decrease in tissue Fraction of Inspired Oxygen (FiO$_2$) catalyzes the accumulation of Hypoxia-Inducible Factor 1 (HIF-1) [67–69], a transcription factor that decays rapidly in normoxic environments [70]. This transcription factor has been validated to effectively activate regulatory factors related to the aforementioned metabolic adaptations [71, 72]. Furthermore, after high-intensity hypoxic training, various adaptation mechanisms closely related to aerobic performance (such as myoglobin concentration, citrate synthase, exercise economy, etc.) have been widely reported [9, 59, 73, 74]. Moreover, several studies have enhanced the comprehension of these physiological adaptations following l-IHT by evaluating related mRNA [9, 72, 75]. However, no studies have directly explored the impact of specific training variables (intensity and volume) on the amplification of aerobic performance in athletes during hypoxic training.

Beyond training intensity and volume, the degree of hypoxia is also a significantly vital element of training prescription. Studies on acute hypoxia have evidenced that exposure to moderate altitudes significantly impairs SaO$_2$ and disturb dynamic balance [76]. The degree of stimulation increases with increasing altitude [77]. However, Karayigit et al. (2022) conducted separate investigations into the acute impacts of moderate and high hypoxia levels on the high-intensity intermittent performance of athletes. The studies revealed no substantial differences between both conditions, and no significant detriments were observed compared with that observed in the normoxic group. Additionally, Warnier et al.'s research found no notable variation in performance enhancement (measured according to incremental test results) across the 2000 m, 3000 m, and 4000 m hypoxic groups following a six-week course of hypoxia training [78]. However, given the scarcity of studies and small sample size, these findings are insufficient to conclusively assert any uniform impact of varied hypoxia levels in high-intensity training on athletes' aerobic capacity. Finally, the physiological responses and adaptations post-hypoxia demonstrate discernible individual variations, which cannot be ignored [53]. Several studies have indicated that elite athletes, compared with athletes at the lower training levels, endure more significant damage in hypoxic conditions, primarily due to the restrictions in pulmonary gas exchange [52]. This finding is corroborated by the strong correlation

between sea-level and hypoxic VO$_2$max [79]. While the oxidative capacity of elite athletes may have reached its limit after extensive years of training, Ponsot et al. reported that post-IHT, the skeletal muscles of high-level athletes exhibit qualitative adaptation (increased Km for ADP) rather than a quantitative one [58]. These adaptations can contribute to improved endurance performance through the better integration of energy demand and utilization. This re-emphasizes that, especially for elite athletes with extended years of training experience, sufficient duration of work intervals is the key to enhancing aerobic performance.

This is the first study attempting to explore the impact of various LLTH variants on athletes' anaerobic performance using meta-analysis. The results show that all active intermittent hypoxic modes compared with normoxic training can effectively improve athletes' anaerobic performance. Further, Although the results demonstrated that hypoxic training with high-intensity and long-duration working interval (l-IHT) has the best improvement effect on anaerobic performance [19], it is regrettable that this study only included one related research, which greatly reduced the statistical power and result credibility. As highlighted, athletes striving to match the same external work rate (absolute exercise intensity) under hypoxic conditions, equivalent to normoxic conditions, require the considerate use of anaerobic metabolism for energy. The training intensity executed directly correlates with the stimulation of anaerobic metabolism—the higher the intensity, the greater the stimulation, and the longer the duration, the more profound the buffering capacity stimulation. Studies on IHT indicate significant correlation between enhancement of anaerobic performance and adaptation of glycolytic enzyme capacity (phosphofructokinase) [63, 65, 80], MCT-1/4 [23], and capillary density. Moreover, the current literature substantiates that moderate acute hypoxia does not curtail athletes' anaerobic function [81]. A noticeable but substantial reduction in resting SaO$_2$, from 95% to 92%, occurs when athletes train above sea level (ambient PIO$_2$ = 159 mmHg) to a moderate altitude (3000 m, ambient PIO$_2$ = 110 mmHg). The situation intensifies at higher altitudes (5000 m, ambient PIO$_2$ = 85 mmHg), where SaO$_2$ plummets to 80% or lower [32]. The significant decrease in SaO$_2$ implies greater demand for and stimulation of anaerobic metabolism. Nevertheless, caution is needed as the more severe the hypoxia, the greater the interference with autonomic nervous system activity [82, 83], leading to the accumulation of fatigue and stress. Therefore, the real-time monitoring of physiological characteristics and training intensity is vital in ensuring the successful execution of hypoxia training.

Czuba et al. (2017) highlighted the essential role of supplementary strength training in enhancing anaerobic performance. Some studies have suggested that hypoxic exposure can have harmful effects on muscle tissue, reducing protein synthesis [54, 84, 85] and leading to muscle fiber atrophy. However, short-term hypoxic exposure, particularly when combined with resistance training, could have a reverse effect through the stimulation of muscle protein synthesis [86, 87], which is beneficial for improving anaerobic performance.

Interestingly, the combination of both continuous and intermittent training in a hypoxic condition has not shown significant enhancement in either aerobic or anaerobic performances. This type of combined hypoxia regimen can be divided into two as follows: 1) continuous and intermittent training sessions that are performed separately each week [22, 88, 89] and 2) continuous and intermittent training conducted during a session [21, 23, 90, 91]. The result of comprehensive data indicates that engaging in additional continuous low-intensity training under hypoxia will not produce additional effects and may even deepen fatigue. The significant effects shown by some combined hypoxia programs may also be due to the contribution of high-intensity intermittent training sessions. Pure passive hypoxic exposure compared with IHE has not shown any enhanced effect on athletic performance, which is consistent with most reviews. Our recent study also proves that IHE cannot improve the maximum oxygen uptake of athletes [15, 30, 92, 93]. Finally, it is worth mentioning that, apart from

l-IHT, RSH is the only hypoxic technique that can simultaneously improve aerobic and anaerobic performances (though marginally). Compared with intermittent hypoxic training, RSH has shorter but more intense work intervals with insufficient recovery in between workouts [28]. This insufficiency stimulates special physiological reactions among athletes after an RSH intervention, with type II muscle fibers displaying greater degree of recruitment and oxygenation capabilities [28, 34]. The correlation between oxidative tendencies (a non-hematological central and peripheral mechanisms) in fast muscle fibers and the enhancement in aerobic and anaerobic performance presents a noticeable trend [17, 94, 95]. Nonetheless, several studies emphatically assert that athletes' VO2max and endurance test results did not improve subsequent to RSH [34, 96–98], but it is noteworthy that Galvin et al. proposed that the amplifying impact of RSH on anaerobic or aerobic performance is closely related to the work-rest ratio [17]. Nevertheless, the marginal enhancement of both aerobic and anaerobic performance may merely constitute supplementary advantages of RSH. In practical application, the primary aim of RSH is to bolster the repeated sprint ability (mixed-oxide metabolism) of team or racquet-sports athletes [28, 34, 92].

## Limitations

While this study confirms an impact of LLTH on aerobic and anaerobic performance based on the intensity, volume, and type of hypoxic training, several confounding factors—unquantifiable in this study—might affect the study outcomes. These include the degree of hypoxia, the integration of additional normoxic training, training frequency, and overall session volume. The effect of these particular arrangements on real-world applications is significant and necessitates further investigation. Furthermore, the characteristics of the research population are specific to males, as they were the majority (92.69%). Some studies have postulated that females exhibit lower $SaO_2$ sensitivity to hypoxic stimuli than that observed in males [99, 100], insinuating minimal impact of hypoxic training on female athletes' performance. Consequently, we cannot definitively determine whether our conclusions are applicable to female athletes. Although some reports have suggested that female athletes can benefit from hypoxic training, the overall sample size is substantially small, and the statistical power of the results is relatively weak. Besides gender, other characteristics such as the sports athletes participate in and their competitive levels are also worthy of further investigation. Unfortunately, the current number of studies included does not support conducting methods like subgroup analysis (after grouping by characteristics, the limited number of studies would severely compromise the credibility of the outcomes and the network connectivity of meta-analysis).

Finally, it is important to note that while the results indicated that l-IHT had the most significant impact on enhancing anaerobic performance, these findings should not be regarded as conclusive due to the small number of pertinent studies analyzed. Further research is necessary to investigate the effects of interval hypoxic training with longer work durations on anaerobic performance, which is vital for the effective practical implementation of hypoxic technique.

## Conclusion

Among the various LLTH strategies, only active intermittent hypoxic modes compared with normoxic training have shown significant enhancements of athletic performance. Intermittent hypoxic training with adequate work-interval durations demonstrated the most advantageous effects on aerobic performances. Neither IHE nor CHT was proven effective.

## Supporting information

**S1 Checklist. PRISMA checklist.**
(DOCX)

**S1 Table. Search strategy.**
(DOCX)

**S2 Table. Physiotherapy evidence database (PEDro) scores of the included studies.**
(DOCX)

**S3 Table. The included studies.**
(DOCX)

**S1 File. Protocol.**
(DOCX)

**S2 File. Classification (figure) and definition (table) of various LLTH modes.**
(DOCX)

**S3 File. Hypoxic dose model.**
(DOCX)

**S4 File. Selection of reference indicators.**
(DOCX)

**S5 File. Statistical methods in details.**
(DOCX)

**S6 File. Evaluation of heterogeneity and inconsistency.**
(DOCX)

## Acknowledgments

We are very grateful to Ms. Fengyu Shi, Mr. Bowen Li and Mr. Yuze Li for their support in data verifcation.

## Author Contributions

**Conceptualization:** Xinmiao Feng, Linin Zhao.

**Data curation:** Xinmiao Feng, Yonghui Chen, Hongyuan Lu, Chuangang Wang.

**Formal analysis:** Xinmiao Feng.

**Funding acquisition:** Xinmiao Feng.

**Investigation:** Xinmiao Feng.

**Methodology:** Xinmiao Feng.

**Project administration:** Xinmiao Feng.

**Resources:** Xinmiao Feng.

**Software:** Xinmiao Feng, Teishuai Yan.

**Supervision:** Xinmiao Feng.

**Validation:** Xinmiao Feng.

**Visualization:** Xinmiao Feng.

**Writing – original draft:** Xinmiao Feng.

**Writing – review & editing:** Xinmiao Feng.

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
