## [Decision Letter · Decision Letter 0]

7 Nov 2023

PONE-D-23-31187Effects of various living-low and training-high modes with distinct training prescriptions on sea-level performance: a network meta-analysisPLOS ONE

Dear Dr. linlin,

Thank you for submitting your manuscript to PLOS ONE. 3 independent reviewers evaluated your manuscript and provided a thorough review of your work. After careful consideration, we feel that it has merit but does not fully meet PLOS ONE’s publication criteria as it currently stands. Therefore, we invite you to submit a revised version of the manuscript that addresses the points raised during the review process.

We look forward to receiving your revised manuscript.

Kind regards,

Raphael Faiss

Academic Editor

PLOS ONE

Journal Requirements:

Reviewers' comments:

Reviewer's Responses to Questions

**Comments to the Author**

1. Is the manuscript technically sound, and do the data support the conclusions?

Reviewer #1: Yes

Reviewer #2: Partly

Reviewer #3: Yes

2. Has the statistical analysis been performed appropriately and rigorously? 

Reviewer #1: Yes

Reviewer #2: Yes

Reviewer #3: Yes

3. Have the authors made all data underlying the findings in their manuscript fully available?

Reviewer #1: Yes

Reviewer #2: Yes

Reviewer #3: Yes

4. Is the manuscript presented in an intelligible fashion and written in standard English?

Reviewer #1: Yes

Reviewer #2: Yes

Reviewer #3: Yes

5. Review Comments to the Author

Reviewer #1: The manuscript is very interesting and well written. All comments and suggestions I provide below have the only purpose to increase the quality of the present work. I hope you find my suggestions useful.

Introduction

Line 43-44: “most studies have focused …………..rest periods”. Authors cited only one reference. Please give more references.

Line 56-58: I suggest authors to cite some of these studies.

Line 68: Please check the spelling of “hypoxic” .

Line 74: Change “Training of varying intensities and durations” to “training at different duration and intensities”.

Line 89: Please add “i.e., aerobic and anaerobic”.

Methods

Line 92: Change “review” to “meta-analysis” since it is a network meta-analysis.

Line 93-94: Change “This study has been registered” to “The study protocol has been registered”.

Line 98-101: In your search terms you have not included "performance" which is in your main concern. How you justify this?

Line 102: Remove “Additionally”.

Line 107: Change “the past 6 months” to the last 6 months”.

Line 124-126: Please add “i.e., cold”.

Line 126: Please add “i.e., nitrate, caffeine”.

Line164-166: I suggest authors to add the reference about the tau squared (τ2).

Line 456-457: Please clarify this sentence “Neither passive hypoxic exposure (IHE) nor moderate-intensity CHT was proven effective”. The abbreviation IHE was defined previously as intermittent hypoxic exposure. If author use “IHE” as an example please add e.g., before.

Tables

Table 1: Please add PEDro score of the included studies.

Table 1: Some abbreviations included in this table are not defined at the foot of the table (e.g., N, H, UCT). Please revise this.

Table 1: Author used the abbreviation is “C+T”, should this be “C+I”?

Reviewer #2: Thank you for providing me with the opportunity to review this study. This meta-analysis, which evaluates the effects of Living low training high across different exercise durations, presents very intriguing findings. Please address the following comments.

Major Comments:

1. The manuscript is dense and represents a robust network meta-analysis; however, please reduce redundancy and ensure that essential information is included in the main text. For instance, while the Introduction contains some details unrelated to the current study's methods, key methodological points are written in the Supplementary File. Also, organize the information from the Supplemental section coherently within the main text. Details on the Supplemental comments are noted in the minor comments section.

2. Abbreviations like RSH and IHT seem to be cited from previous studies; presenting them as a list makes it difficult to understand due to inconsistent use of initialisms and terminology, some including 'Hypoxia' and others 'Training'. I strongly recommend revising for clarity. Additionally, as these definitions are crucial to the paper, include them in the main text rather than in the supplemental material.

3. There is only one study on the improvement of anaerobic performance by L-IHT (Table 1), so it is questionable to treat this as a main finding.

4. Most studies on L-IHT involve endurance athletes. Please mention how the type of training modality may influence the characteristics of the athlete in the discussion, or at the very least, as a limitation.

5. Please add a discussion on the mechanism by which RSH may improve aerobic performance.

Minor Comments:

1. In the Abstract, please describe the duration of exercise for each exercise modality.

2. Line 68, remove the unnecessary spaces.

3. Lines 80-83, since the study focuses on performance, there is no need to discuss health.

4. Line 163, please format “squared” as a superscript.

5. Specify which performance indicators were used in the main text and appropriately cite Table S6.

6. Since Supplement S5 is short, include it in the main text.

7. In Table 1, use 'sports' instead of 'subject', and ensure uniformity by using the names of the sports (i.e., swimming) instead of participant types (i.e., swimmers).

8. In Table 1, the Hypoxic protocol is also mentioned in S8. Please delete S8.

9. In Table 1, it is Kasai et al., not Nobukazu et al. (Supplemental reference 33), and the participants were lacrosse players, not cyclists.

Reviewer #3: Effects of various living-low and training-high modes with distinct training prescriptions

on sea-level performance: a network meta-analysis:

First of all, the reviewer would like to thank the authors for their work and efforts in trying to improve sports science knowledge. The article is an interesting approach to the ffects of various living-low and training-high modes with distinct training prescriptions on sea-level performance. The study is well designed and well-written, with a great introduction proposing the usefulness of the topic and a clear outline of the research question. I suggest that the this article can be accepted without any revisions.

6. PLOS authors have the option to publish the peer review history of their article (what does this mean?). If published, this will include your full peer review and any attached files.

Reviewer #1: **Yes: **Fatma Rhibi

Reviewer #2: **Yes: **Daichi Yamashita

Reviewer #3: No

---

## [Author Response · Author response to Decision Letter 0]

22 Nov 2023

Dear Reviewers,

I sincerely appreciate the time and effort you have taken to review and provide insightful feedback on my manuscript titled “[Effects of various living-low and training-high modes with distinct training prescriptions on sea-level performance: a network meta-analysis]”. Your valuable comments have provided a clear direction towards improving my research work.

Each of your suggestions has been taken into consideration and applied to revise the manuscript. Your insights triggered a rethinking process and the manuscript has undergone substantial revision. The content has been restructured and details have been added as necessary to address the concerns you’ve raised. The methodological and conceptual changes you suggested have significantly enhanced the clarity and quality of our research.

Your expertise and unequivocal clarity are commendable, and your inputs have enriched my manuscript. I believe it will be more engaging and impactful after having integrated your suggestions.

Thank you once again for your thoughtful and enlightening perspectives. It has been a valuable learning experience, and I look forward to more opportunities to learn from your expertise in the future.

Best regards,

Xinmiao Feng

Beijing sports university

---

## [Editor Report · Decision Letter 1]

23 Nov 2023

PONE-D-23-31187R1Effects of various living-low and training-high modes with distinct training prescriptions on sea-level performance: a network meta-analysisPLOS ONE

Dear Dr. linlin,

Thank you for submitting your manuscript to PLOS ONE. After careful consideration, we feel that it has merit but does not fully meet PLOS ONE’s publication criteria as it currently stands. Therefore, we invite you to submit a revised version of the manuscript that addresses the points raised during the review process.

Dear Author,

Thank you for your hard work in submitting a revision of the manuscript.

After initial review by myself and before submitting to the reviewers after your answers to their comment, i noticed a samll error in the "labelling" of the studies included (Table 2).

For several studies, you inversed first name and last name of the corresponding author.

Please correct for

Paul et a.

Margaux et al.

David et al.

Raphael et al.

Harvey et al.

Hannes et al.

Please note also that for the Faiss et al (2013) study with cyclists (labeled here as "Raphael et al.) , there were 20 subjects for N and 20 for H (not 25!) since there was a control group with no training of 10 subjects.

Please correct also the page numbering in the uploaded file.

Once you have been able to process the file (with track changes and in original format), i will forward it to the reviewers for their evaluation.

Thank you in advance,

Raphael Faiss

We look forward to receiving your revised manuscript.

Kind regards,

Raphael Faiss

Academic Editor

PLOS ONE
---

## [Author Response · Author response to Decision Letter 1]

25 Nov 2023

dear Faiss:

It is an honor for us to receive your letter. Your research in the field of sports has provided us with tremendous assistance. On behalf of my co-authors, we thank you for giving us a chance to revise and improve the quality of our article. Compared to our previous submission experiences, what impressed us is that the peer review process at PLOS ONE is so meticulous and rigorous. Every member of our team admires your work and we also appreciate your valuable suggestions. Finally, we would like to apologize to you as our team’s negligence has increased your workload.

(1)After initial review by myself and before submitting to the reviewers after your answers to their comment, i noticed a samll error in the "labelling" of the studies included (Table 2).

For several studies, you inversed first name and last name of the corresponding author.

Please correct for 

Paul et a.

Margaux et al.

David et al.

Raphael et al.

Harvey et al.

Hannes et al.

Please note also that for the Faiss et al (2013) study with cyclists (labeled here as "Raphael et al.) , there were 20 subjects for N and 20 for H (not 25!) since there was a control group with no training of 10 subjects.

Response:

The aforementioned problem were due to the negligence of the team member responsible for the extraction of literature characteristics of RHS intervention. Firstly, we have rectified the error of "labelling" in Table 2. Secondly, we wholeheartedly thank you for pointing out the erroneous subjects number in this study conducted by Faiss et al. This error has been rectified in the abstract, results, Table 2, limitations, and supported information. Lastly, it is essential to acknowledge that the error in the inclusion subjects number could potentially impact the statistical results. Therefore, we have conducted a comprehensive review of all the data to address this issue. Fortunately, the modification made to the inclusion number in the study did not result in any significant deviation in the research findings (the comparison of former results and modified results has been show in the new "respond to reviewer" dox.). All of the above modification have been marked by red in the revised manuscript. These errors and defects are actually quite serious, but we are extremely grateful for the opportunity you have given us.

XM feng

Beijing

2023.11.25

---

## [Decision Letter · Decision Letter 2]

19 Dec 2023

PONE-D-23-31187R2Effects of various living-low and training-high modes with distinct training prescriptions on sea-level performance: a network meta-analysisPLOS ONE

Dear Dr. linlin,

Thank you for submitting your manuscript to PLOS ONE.Reviewers indicate that you have addressed most of their previous comments. At this stage, only minor amendments need to be done for further consideration in the publication process (See point 6 below)Therefore, we invite you to submit a revised version of the manuscript that addresses the last few points raised during the review process.

We look forward to receiving your revised manuscript.

Kind regards,

Raphael Faiss

Academic Editor

PLOS ONE

Journal Requirements:

Reviewers' comments:

Reviewer's Responses to Questions

**Comments to the Author**

1. If the authors have adequately addressed your comments raised in a previous round of review and you feel that this manuscript is now acceptable for publication, you may indicate that here to bypass the “Comments to the Author” section, enter your conflict of interest statement in the “Confidential to Editor” section, and submit your "Accept" recommendation.

Reviewer #1: All comments have been addressed

Reviewer #2: All comments have been addressed

2. Is the manuscript technically sound, and do the data support the conclusions?

Reviewer #1: Yes

Reviewer #2: Yes

3. Has the statistical analysis been performed appropriately and rigorously? 

Reviewer #1: Yes

Reviewer #2: Yes

4. Have the authors made all data underlying the findings in their manuscript fully available?

Reviewer #1: Yes

Reviewer #2: Yes

5. Is the manuscript presented in an intelligible fashion and written in standard English?

Reviewer #1: Yes

Reviewer #2: Yes

6. Review Comments to the Author

Reviewer #1: Authors have adequately addressed my comments raised in a previous round of review.

I feel that this manuscript is now acceptable for publication.

Reviewer #2: The manuscript has shown improvement, and I believe it meets the standards for publication in PLOSONE with a few minor modifications.

Page 3, Lines 56-57: The phrase 'repeated sprint training in hypoxia' is mentioned.

Page 3, Line 58: The term 'interval sprint training in hypoxia' is used.

Page 6, Lines 115-127: There are several concerns in the revised section:

The term “al” is unclear. Could you please clarify what this refers to?

Please format the manuscript properly. Ensure there is a space between numbers and symbols.

The unit notation is ambiguous. The term 's/min' could be interpreted as seconds or minutes, but it currently appears as if it means 'second divided by minute'. Please clarify.

7. PLOS authors have the option to publish the peer review history of their article (what does this mean?). If published, this will include your full peer review and any attached files.

Reviewer #1: **Yes: **Fatma Rhibi

Reviewer #2: **Yes: **Daichi Yamashita

---

## [Author Response · Author response to Decision Letter 2]

20 Dec 2023

On behalf of all the contributing authors, I would like to express our sincere appreciations of editor and reviewers’ constructive comments concerning our article entitled “Effects of various living-low and training-high modes with distinct training prescriptions on sea-level performance: a network meta-analysis”. These comments are all valuable and helpful for improving our article.

Response to Reviewers

Response to Reviewer 1: Authors have adequately addressed my comments raised in a previous round of review. I feel that this manuscript is now acceptable for publication.

Response: thanks for your careful checks and supports.

Response to Reviewer 2: The manuscript has shown improvement, and I believe it meets the standards for publication in PLOSONE with a few minor modifications.

Reviewer #2

We feel great thanks for your professional review work on our article. As you are concerned, there are several problems that need to be addressed. According to your nice suggestions, we have made extensive corrections to our previous draft, the detailed corrections are listed below.

(1)Page 3, Lines 56-57: The phrase 'repeated sprint training in hypoxia' is mentioned.

Response: thanks for your careful checks. We have made a correction to the phrase, at at Line 56 to 57 on page of 3 and already marked in red font.

(2)Page 3, Line 58: The term 'interval sprint training in hypoxia' is used.

Response: thanks for your careful checks. We have made a correction to the phrase, at at Line 58 on page of 3 and already marked in red font.

(3)Page 6, Lines 115-127: There are several concerns in the revised section:

The term “al” is unclear. Could you please clarify what this refers to?

Please format the manuscript properly. Ensure there is a space between numbers and symbols.

The unit notation is ambiguous. The term 's/min' could be interpreted as seconds or minutes, but it currently appears as if it means 'second divided by minute'. Please clarify.

Response: thanks for your careful checks. We are sorry for our carelessness. Based on your comments, we have made the corrections, that “al” is wrong spell, formt and the expression of “s/min” has been modified, at at Line 116 to 128 on page of 6 and already marked in red font.

---

## [Editor Report · Decision Letter 3]

27 Dec 2023

Effects of various living-low and training-high modes with distinct training prescriptions on sea-level performance: a network meta-analysis

PONE-D-23-31187R3

Dear Dr. linlin,

We’re pleased to inform you that your manuscript has been judged scientifically suitable for publication and will be formally accepted for publication once it meets all outstanding technical requirements.

Kind regards,

Raphael Faiss

Academic Editor

PLOS ONE

---

## [Editor Report · Acceptance letter]

28 Mar 2024

PONE-D-23-31187R3 

PLOS ONE

Dear Dr. Zhao, 

I'm pleased to inform you that your manuscript has been deemed suitable for publication in PLOS ONE. Congratulations! Your manuscript is now being handed over to our production team.

Kind regards, 

on behalf of

Dr. Raphael Faiss 

Academic Editor

PLOS ONE